# The Hippocampal Response to Acute Corticosterone Elevation Is Altered in a Mouse Model for Angelman Syndrome

**DOI:** 10.3390/ijms24010303

**Published:** 2022-12-24

**Authors:** Eva M. G. Viho, A. Mattijs Punt, Ben Distel, René Houtman, Jan Kroon, Ype Elgersma, Onno C. Meijer

**Affiliations:** 1Department of Medicine, Division of Endocrinology, Leiden University Medical Center, Albinusdreef 2, 2333 ZA Leiden, The Netherlands; 2Einthoven Laboratory for Experimental Vascular Medicine, Leiden University Medical Center, 2333 ZA Leiden, The Netherlands; 3Department of Clinical Genetics, Erasmus MC, 3015 GD Rotterdam, The Netherlands; 4ENCORE Expertise Center for Neurodevelopmental Disorders, Erasmus MC, 3015 GD Rotterdam, The Netherlands; 5Precision Medicine Lab, 5349 AB Oss, The Netherlands

**Keywords:** stress, Angelman syndrome, glucocorticoid receptor, ubiquitin-protein ligase E3A, hippocampus

## Abstract

Angelman Syndrome (AS) is a severe neurodevelopmental disorder, caused by the neuronal absence of the ubiquitin protein ligase E3A (UBE3A). UBE3A promotes ubiquitin-mediated protein degradation and functions as a transcriptional coregulator of nuclear hormone receptors, including the glucocorticoid receptor (GR). Previous studies showed anxiety-like behavior and hippocampal-dependent memory disturbances in AS mouse models. Hippocampal GR is an important regulator of the stress response and memory formation, and we therefore investigated whether the absence of UBE3A in AS mice disrupted GR signaling in the hippocampus. We first established a strong cortisol-dependent interaction between the GR ligand binding domain and a UBE3A nuclear receptor box in a high-throughput interaction screen. In vivo, we found that UBE3A-deficient AS mice displayed significantly more variation in circulating corticosterone levels throughout the day compared to wildtypes (WT), with low to undetectable levels of corticosterone at the trough of the circadian cycle. Additionally, we observed an enhanced transcriptomic response in the AS hippocampus following acute corticosterone treatment. Surprisingly, chronic corticosterone treatment showed less contrast between AS and WT mice in the hippocampus and liver transcriptomic responses. This suggests that UBE3A limits the acute stimulation of GR signaling, likely as a member of the GR transcriptional complex. Altogether, these data indicate that AS mice are more sensitive to acute glucocorticoid exposure in the brain compared to WT mice. This suggests that stress responsiveness is altered in AS which could lead to anxiety symptoms.

## 1. Introduction

Angelman Syndrome (AS) is a neurodevelopmental disorder characterized by severe intellectual disability, motor deficits, anxiety, speech impairments, ataxia, sleep disturbances, and epilepsy [1,2,3]. AS is caused by the deficiency or loss-of-function of the ubiquitin protein ligase E3A (UBE3A) in neurons [4,5]. The *UBE3A* gene is located on chromosome 15 in a region (15q11.2–13.1) that is subjected to neuron– and paternal–specific genomic imprinting, making the maternal allele of *UBE3A* the sole source of UBE3A protein in mature neurons. Chromosomal aberrations such as deletions, uniparental disomy, imprinting defects, or point mutations that affect the integrity of the maternal *UBE3A* gene lead to the complete loss of neuronal UBE3A, resulting in AS [6,7,8,9,10]. UBE3A is an E3-ligase involved in the ubiquitin-proteasome cascade, a system required for the breakdown and recycling of redundant or defective proteins [11,12,13,14]. Mutations affecting UBE3A activity or its nuclear localization lead to AS, suggesting that UBE3A plays an important role in the nucleus [15,16,17,18]. Notably, UBE3A contains short leucine-rich (LxxLL) binding motifs, also known as nuclear receptor boxes (NR-boxes) and has been reported to be a transcriptional coregulator of nuclear hormone receptors, likely independent of its ligase function [19,20,21,22]. Since loss of UBE3A ligase activity leads to AS [15,16], the exact contribution of UBE3A as a ligase-independent transcriptional coregulator in the context of AS pathophysiology remains understudied.

The glucocorticoid receptor (GR) is a ligand-dependent transcription factor that plays a critical role in stress response and subsequent adaptation [23,24,25,26]. The GR is activated by glucocorticoid hormones, such as cortisol which is predominant in humans and corticosterone which is exclusive in mice. Glucocorticoid release into the circulation is controlled by the hypothalamic pituitary adrenal (HPA) axis [27] and follows a circadian rhythm with high hormone levels at the beginning of the active phase and lower levels during the resting phase [28,29]. Upon activation, the GR changes conformation resulting in its nuclear translocation. In the nucleus, the GR interacts with other transcription factors or directly binds to palindromic DNA sequences called glucocorticoid-response elements (GRE) and recruits coregulator proteins to regulate gene expression [30]. The availability and recruitment of GR coregulators is highly context-specific, which leads to a large diversity in GR-driven transcriptomes throughout the brain and between cell types within one brain region including the hippocampus [31,32]. Disrupted GR signaling in the hippocampus of UBE3A-deficient AS mice has been associated with learning disabilities and anxiety-like behavior [33]. Furthermore, UBE3A-deficient AS mice are known to be hypersensitive to diet-induced liver steatosis [34], and previously showed metabolic disturbances resulting from an altered circadian rhythm [35]; characteristics that have been related to disruptions in GR signaling [36,37,38]. However, the molecular mechanisms underlying the UBE3A modulation of GR signaling are not fully understood. Here, we aimed to better understand the interaction between UBE3A and GR, including the transcriptomic effects of acute or chronic corticosterone treatment on GR signaling in the hippocampus and liver tissue of WT and UBE3A-deficient AS mice.

## 2. Results

### 2.1. UBE3A Interacts with the GR Ligand Binding Domain

The identification of GR coregulatory interactions is essential in the understanding of GR signaling adaptive and maladaptive outcomes. We evaluated the interaction between the GR LBD with an array of known nuclear receptor coregulator binding motifs in the presence of 1 µM cortisol. The short leucine-rich (LxxLL) binding motifs, also known as nuclear receptor boxes (NR-boxes), of 64 known nuclear receptor coregulators were incubated with the GR LBD that was tagged with Glutathione S-transferase (GST). The binding of GR LBD to a coregulator NR-box was detected with an anti-GST antibody coupled to a fluorophore. The modulation index (MI) represents the normalized immunofluorescent signal in the presence of cortisol (Figure 1A). Out of 154 NR-boxes, 54 showed significant interaction with GR LBD in the presence of 1 µM cortisol, as compared to vehicle (Figure 1B). In this subset, we could identify NR-boxes from nuclear receptor coactivators including the SRC family (NCOA1, NCOA2, NCOA3 and NCOA6), PRGC1, and known corepressors such as LCOR and NRIP1 (Figure 1B and Appendix A). The strongest interaction with GR LBD was observed for one NR-box of UBE3A (Figure 1B). This LxxLL motif of UBE3A localized between amino acid 393 and 415 (MI = 1.89, p = 0.03) of the human isoform 3 (UBE3A hIso3, [39]). Another significant interaction was found between GR LBD and a UBE3A hIso3 LxxLL motif situated between amino acid 646 and 668 (MI = 0.55, *p* = 0.004) (Appendix A). We further investigated the effect of UBE3A on GR activity and stability in cultured cells. Here, we found that GR activity in response to cortisol was significantly reduced in UBE3A^KO^ HEK 293T cells, suggesting that UBE3A can act as a GR transcriptional coregulator in these cells (Figure 1C). Since GR transcriptional activity was previously linked to its proteasomal degradation, we assessed whether UBE3A enhanced the ubiquitin-dependent degradation of cortisol-bound GR. In contrast with the ligase-dead mutant (UBE3A^LD^), active nuclear (UBE3A^WT^) and cytosolic UBE3A (UBE3A^dAZUL^) modestly—but non-significantly—reduced GR levels, as compared to the empty vector (EMPTY). These modest effects could be rescued by inhibiting the proteasome (Figure 1D). However, GR activation by cortisol did not further destabilize GR in the presence of active UBE3A (Figure 1E). Additionally, we did not observe UBE3A-dependent enhancement of GR turnover following the addition of de novo protein synthesis inhibitor CHX (Figure 1F). Taken together, these results indicate that it is unlikely that the GR coregulator function of UBE3A relies on the proteasomal degradation of cortisol-bound GR, and rather support previous findings suggesting this process is largely independent of UBE3A ligase activity. 

### 2.2. UBE3A Deficiency in AS Mice Alters Circulating Corticosterone Levels

We next evaluated the role of UBE3A in GR signaling in vivo. To assess whether UBE3A deficiency in AS mice could interfere with HPA axis activity and alter circadian glucocorticoid release, we measured baseline plasma corticosterone (CORT) levels in UBE3A-deficient AS mice and WT littermates. As expected, WT and AS mice showed significantly higher plasma CORT levels at the beginning of their active phase (PM) as compared to the resting phase (AM) (*p* < 0.001, Figure 2A). Remarkably, AS mice showed low to undetectable levels of CORT at AM, even though at PM the levels were comparable to WT. This makes the daily corticosterone fluctuations in AS mice significantly larger than in WT mice (Figure 2B), which indicates that HPA axis activity is altered in AS mice. 

### 2.3. Acute CORT Exposure Strongly Alters Hippocampal GR Signaling in UBE3A-Deficient AS Mice 

Alterations of HPA axis activity can lead to further disruption of GR signaling in response to abnormal increases in glucocorticoid levels. We compared the hippocampus transcriptomic response in UBE3A deficient adult AS mice and WT littermates in response to acute CORT exposure, using RNA-seq (Appendix A). Hereto, UBE3A-deficient AS mice and WT mice were subcutaneously injected with 3 mg/kg CORT or vehicle, three hours prior to brain tissue collection (Figure 3A). We confirmed the lack of UBE3A protein expression in brain tissue of AS mice (*p* < 0.001) and found that UBE3A levels were not influenced by acute CORT treatment (Figure 3B and Appendix A). The PCA analysis on the RNA-seq data revealed that 36% of the variation (PC1 + PC3) was explained by the genotype (AS vs. WT) and treatment (CORT vs. vehicle) effects. The transcriptomic profiles of WT and AS mice largely overlapped upon vehicle treatment but diverged after CORT treatment (Figure 3C). PC2 corresponded to 18% of the variation and was mostly explained by a single WT mouse, treated with CORT (i.e., sample H8; Appendix A). This mouse did not display abnormal plasma CORT levels at endpoint (Appendix A), hence this sample was included in the subsequent analysis. Upon acute CORT treatment (FDR-adjusted *p*-value < 0.05), the expression of 283 genes was altered in the hippocampus of WT mice (Figure 3D), as compared to 2910 genes in AS mice (Figure 3E). Among these genes differentially regulated by CORT, 204 overlapped between WT and AS, 79 were WT-specific, and 2706 were AS-specific (Figure 3D,E). These results can partially be explained by the slightly elevated GR levels measured in AS brain tissue (*p* = 0.0594, Figure 3F and Appendix A), but the magnitude of the transcriptomic effects suggests that UBE3A exerts a limiting effect on GR transcriptional activity in the mouse hippocampus. 

### 2.4. Acute CORT Exposure Influences Pathways Associated with Transcription Activity and Neurotransmitter Signaling in the Hippocampus of UBE3A-Deficient AS Mice

When comparing the CORT treated WT group with the CORT treated AS group, we found 3,890 differentially regulated genes (Appendix A). Of these, 1,208 genes statistically contributed to the interaction between the genotype and the treatment, and therefore underly the differential response to acute CORT in AS mice (Appendix A). We performed gene ontology and pathway enrichment analyses on that gene subset and the top 10 GO-terms for molecular functions were associated with amino acid transmembrane transport (solute carrier family—SLC) and transcription activity, including nuclear hormone receptor activity (Figure 4A). This last term included 21 genes regulated by CORT in AS mice (Appendix A), including genes involved in the mediator complex (*Med24*, *Med4*), transcription coactivators (*Snw1*, *Tgfb1i1*, *Wbp2*, *Thrap3*), chromatin remodelers (*Smarce1*, *Hmga1b*, *Sirt1*, *Bcas3*), transcription factors (*Tcf7l2*, *Lef1*, *Nr1h2*, *Tcf7l2*), proteins involved in nuclear receptor import (*Ipo13*) or protein stability (*Sumo2*, *Rnf6*). The top 10 enriched pathways according to the REACTOME database were predominantly involved in mRNA splicing, chromatid cohesion (*Pds5b*, *Rad21*, *Esco1*, *Smc3*, *Wapal*, *Stag1*, *Stag2*) and neurotransmitter signaling (*Ppfia2*, *Ppfia3*, *Stx1a*, *Vamp2*, *Lin7c*, *Slc6a11*, *Slc17a7*, *Cplx1*, *Stxbp1*, *Rab3a*, *Syn1*) (Figure 4B and Appendix A). This suggests that the effects of acute CORT exposure in the brains of AS, but not WT mice are broadly involved in transcription regulation and synaptic transmission.

### 2.5. Acute CORT Exposure Influences Genes in the UBE3A-Deficient AS Mouse Hippocampus That are Heterogeneously Expressed in Hippocampal Cell Types

We selected the top 40 genes from the 1,208 genes which significantly contributed to the genotype-treatment interaction and characterized their cell type-specific expression using our adult mouse hippocampus atlas (Figure 4C). Two genes from the selected subset were undetectable in the hippocampus cell atlas: *Mir5125* which was not captured by 10× scRNA-seq, and *Adnp*. The expression of genes appeared heterogenous throughout different hippocampal cell types, e.g., *Rfx3* and *Rasal2* were highly expressed in the glutamatergic neurons of the dentate gyrus (DG), while Ccdc43 and Prpf33a expression was higher in CA2 glutamatergic neurons, and *Ezr* expression was specific for astrocytes (Figure 4C). It is interesting to note that many genes that were downregulated by CORT in AS mice were predominantly expressed in non-neuronal cells, considering that these cells still express 50% of UBE3A protein. For example, *Sox10* was specifically expressed in oligodendrocytes, while *Fxyd5* and *Lef1* were exclusively expressed in endothelial cells. We observed only few genes that were highly expressed in neuronal cell types, for instance *Atf5* and *Lypla2* (Figure 4C). Altogether, these results suggest that genes affected by acute CORT exposure in AS are heterogeneously expressed in mouse hippocampal cell types and sub-regions (DG, CA1-ProS, CA2 and CA3). Each of these cell types may respond differently to reductions in UBE3A levels, thereby featuring specific GR-mediated adaptations to CORT exposure.

### 2.6. Continuous CORT Exposure Does Not Differentially Alter Hippocampal GR Signaling in UBE3A-Deficient AS Mice 

The detrimental effects of GR disruption in the brain are mostly associated with chronic stress or chronic glucocorticoid exposure. Therefore, we investigated whether continuous CORT treatment of AS mice led to similar or even more pronounced changes in hippocampal GR signaling than the acute treatment. AS and WT mice were implanted subcutaneously with a vehicle pellet or a CORT-releasing pellet, such that CORT was continuously released during five consecutive days (Figure 5A). We again confirmed that UBE3A levels were significantly lower in AS mice brain, as compared to WT (*p* < 0.001), and that continuous CORT treatment did not influence UBE3A protein expression in the frontal cortex (Figure 5B and Appendix A). 

Two technical outliers were detected (i.e., samples H9 and H13, Appendix A) and excluded from the RNA-seq analysis. PCA results showed that 43% of the variation (PC1 + PC2) was attributed to the genotype and treatment effects. Similar to what was observed upon acute CORT treatment, continuous CORT treatment resulted in a clear separation of treated and untreated mice. Vehicle-treated WT and AS mice largely overlapped, and CORT treatment resulted in a modest divergence of WT and AS treated animals (Figure 5C). In response to continuous CORT, the expression of 610 genes was altered in the hippocampus of WT mice (Figure 5D), compared to 773 genes in AS (Figure 5E). Among these CORT-regulated genes, 322 genes overlapped between WT and AS, 288 genes were WT-specific, and 451 genes were AS-specific (Figure 5D,E). The differences between AS and WT in response to continuous CORT exposure were less pronounced than after acute treatment (Figure 3D,E). When comparing the CORT treated WT and AS mice, we found only 44 differentially regulated genes, of which none contributed significantly to the genotype-treatment interaction (Appendix A and Appendix A), suggesting that the overall transcriptomic effects of continuous CORT are comparable in WT and AS mouse hippocampus. 

Following five days of CORT exposure, we found that the GR levels were slightly, but non significantly, higher in AS brains (*p* = 0.058, Figure 5F). A much larger effect on GR abundance was seen after CORT treatment, which strongly decreased GR protein levels (*p* < 0.001, Figure 5F and Appendix A). Altogether, the results indicate that UBE3A deficiency in AS mice does not significantly influence the hippocampal transcriptomic response to continuous CORT exposure. 

### 2.7. Continuous CORT Exposure Slightly Alters Liver GR Signaling in UBE3A-Deficient AS Mice 

AS mice are known to be susceptible to metabolic disturbances and liver steatosis which can be triggered by chronic glucocorticoid treatment, hence we investigated the peripheral effects of continuous CORT treatment in WT and AS mice using RNA-seq on liver tissue (Figure 6A). As expected, UBE3A was expressed in the AS liver at around 50% of WT levels (due to the active paternal allele in this tissue) (genotype effect: *p* < 0.001, Figure 6B and Appendix A). 

According to the PCA results, 48% of the variation (PC1 + PC3) was attributed to the genotype (AS vs. WT) and treatment (CORT vs. vehicle) effects. Comparable to hippocampal data, WT and AS mice showed most overlap upon vehicle treatment and diverged after CORT treatment, which means that most of the effects were driven by CORT treatment (Figure 6C). PC2 corresponded to 23% of the variation and was mostly explained by individual variability between mice (Appendix A). In total, the expression of 2484 genes was altered by continuous CORT treatment in the liver of WT mice (Figure 6D), as compared to 3571 genes in AS mice (Figure 6E). Among these differentially regulated genes, 1811 genes overlapped between WT and AS, 673 genes were WT-specific, and 1762 genes were AS-specific (Figure 6D,E). A total of 91 genes were differentially regulated between CORT treated AS and WT mice (Appendix A), of which only 10 genes statistically contributed to the interaction between the genotype and the treatment (Appendix A), a dataset too small to perform any adequate GO-term and pathway analyses. In the liver of AS mice, GR protein abundance was significantly higher than in WT mice upon vehicle treatment (*p* < 0.05) and continuous CORT exposure led to a significant decrease in GR levels only in AS mice (*p* < 0.01, Figure 6F and Appendix A). Altogether, these results suggest that continuous CORT exposure induces considerable changes in liver transcriptome. Despite the AS-specific decrease in liver GR protein expression upon continuous CORT treatment, only a very small proportion of those transcriptomic effects were specific to AS.

## 3. Discussion

UBE3A deficiency in AS has been previously associated with disrupted GR signaling in the hippocampus and impairments in circadian rhythm [33,35]. In the current study, we confirmed UBE3A interaction with the GR LBD via LxxLL domains and further investigated whether UBE3A-deficient AS mice displayed a differential transcriptomic response to acute and continuous glucocorticoid exposure. 

GR transcriptional activity depends on its interactions with coregulator proteins, which bind to either its N-terminal or its C-terminal coregulator binding domain. C-terminal coregulators, associated with the LBD and AF2, possess leucine-rich motifs (LxxLL) which are required to interact with the GR and determine the transcriptional outcome [30,40]. UBE3A contains three LxxLL motifs [41], two of which showed a significant interaction with the GR LBD. Given the previously reported interaction between full-length UBE3A and GR in mouse brains [33], our results suggest that this binding relies on a physical association between the LBD of GR and these two LxxLL motifs in UBE3A. It is tempting to speculate that the abrogation of this interaction, by mutations in the LxxLL domain, could severely impact the coregulating capability of UBE3A, and contribute to transcriptomic alterations or abnormal behaviors. However, it is difficult to establish how such mutations could specifically affect GR signaling, since UBE3A LxxLL domains have been found to interact with multiple nuclear hormone receptors, including the receptors for estrogen (ERα and ERβ), progesterone (PR), and androgens (AR) [40]. An additional question is whether LxxLL mutations affect the subcellular localization, activity, or stability of UBE3A [16]. If these properties remain largely intact, UBE3A would still be able to function as an E3 ligase for most of its targets, which would prevent LxxLL mutations from causing a typical AS phenotype. These two issues make it difficult to precisely ascertain the pathological contribution of GR dysregulation in AS.

The interaction of UBE3A with GR could precede its ubiquitination and subsequent proteasomal degradation. However, the overexpression of active UBE3A did not significantly decrease GR stability in cells, which indicates that GR is not a typical UBE3A substrate and suggests that the coregulator function of UBE3A may be ligase independent [20,22]. In line with these results, we found that brain GR levels were not significantly altered in AS mice. However, in AS mice livers, we found GR levels to be significantly higher compared to WT. These results are noteworthy considering that UBE3A expression in peripheral tissue of AS mice is substantially higher than in the brain, and as a consequence, one would expect that the effect on GR levels would be less pronounced in the liver. This indicates that tissue-specific targeting mechanisms by UBE3A may play a more prominent role in regulating GR levels, rather than UBE3A dosage. Further experiments are necessary to establish whether GR destabilization originates from direct or indirect ligase-dependent effects of UBE3A.

We established that AS mice displayed almost undetectable levels of CORT during the circadian trough, indicating that their HPA axis activity was influenced by UBE3A deficiency. The circadian dynamics of CORT release in the circulation are centrally regulated in the hypothalamic paraventricular nucleus (PVN), which receives negative feedback via the GR [42]. Previous results on circadian rhythmicity in AS mice demonstrated that specifically in neuronal tissue, the lack of UBE3A compromised the turnover of the circadian clock component BMAL1, which led to impaired rhythmicity [35]. Therefore, the lack of UBE3A in PVN neurons could explain the disruption in diurnal CORT release observed in AS mice. Another factor to consider in the context of the alterations in CORT levels is the mineralocorticoid receptor (MR). The MR has a higher affinity to glucocorticoid hormones as compared to the GR [24], and loss of MR function is known to be associated with memory impairments, increased susceptibility to stress, and anxiety-like behaviors [43,44,45]. However, MR also mediates tonic negative feedback [46], and the differences between AS and WT mice in CORT levels at the circadian trough suggest that MR may be equally or more active in the AS mouse brain. 

Our RNA-seq analysis provided genome-wide evidence of an altered hippocampal transcriptome in AS mice following acute CORT exposure. This treatment affected 10 times as many genes in the AS hippocampus as in WT, which suggests that under normal circumstances UBE3A limits GR signaling in the mouse hippocampus. This is in contradiction with our in vitro results where GR response to cortisol was decreased in UBE3A^KO^ cells, and previous findings in HeLa and Neuro2a cells, where UBE3A acted as a GR transcriptional coactivator [20,33]. The discrepancy between cell models and our in vivo model, warrants further investigation of cell-type specific GR responses, and genomic rather than reporter plasmid readouts to provide more insights into the molecular and cell-type specific events associated with UBE3A-GR interaction. 

The majority of the genes that contributed to the acute CORT differences in the AS mouse hippocampus were associated with transcriptional activity; including transcription factors, coactivators, and chromatin remodelers such as *Smarce1*, which was recently associated with an AS-like phenotype in patients with no molecular diagnostic [47]. Further work is needed to establish how these factors contribute to the differences between AS and WT mice. Besides that, acute CORT treatment in the AS hippocampus also strongly dysregulated genes associated with neurotransmitter signaling. These alterations may disturb synaptic function in CORT-treated AS mice and lead to disruptions in neuronal signaling. Adequate electrophysiological measurements are needed to determine whether CORT exposure leads to measurable alterations in neuronal communication in AS mice. GR is strongly co-expressed with the receptors for serotonin, dopamine, noradrenaline, and acetylcholine in the mouse hippocampal glutamatergic neurons [32], and therefore, disruptions in neurotransmitter signaling could be intertwined with the differential response to CORT in AS mouse hippocampus [48]. There is a multitude of studies that investigated synaptic function in AS mice, collectively reporting cortical and hippocampal neuronal inhibition/excitation imbalances and deficits in hippocampal long-term potentiation [49,50,51,52,53]. Although many synaptic UBE3A substrates have been suggested to effectuate these changes [54,55,56,57], the CORT-induced alterations in neurotransmitter genes we report here could similarly underlie the disrupted electrophysiological properties of AS neurons. Dedicated electrophysiological experiments would be needed to determine whether CORT exposure leads to measurable alterations in synaptic function in AS mice.

When mapping the genes that contributed to the differential transcriptomic effects in the hippocampus of AS and WT mice to our scRNA-seq database, we found that their expression was not restricted to neurons. Several corticosterone-responsive genes were also expressed—sometimes exclusively—in astrocytes, oligodendrocytes, endothelial cells, or microglia. In contrast with AS neurons that are devoid of UBE3A protein, non-neuronal AS cells like astrocytes and oligodendrocytes do not undergo epigenetic silencing of the paternal allele of *Ube3a* during maturation and thus still express UBE3A [58]. Our results indicate that, although 50% of UBE3A protein was maintained, acute CORT treatment also affected the transcriptome of non-neuronal AS cells. These findings emphasize that the contribution of non-neuronal cells should be further explored to understand whether the effects in these cells are the result of a decrease in the molecular interaction between UBE3A and GR. Alternatively, the absence of UBE3A in neurons could influence non-neuronal cells via paracrine signaling. 

In sharp contrast with our findings after acute treatment, no significant genotype-treatment interaction was found in the hippocampus after continuous CORT treatment. This confirms the differences in GR response to acute and continuous CORT treatment in the brain, as expected based on previous findings with acute and chronic stress paradigms [59,60]. Our results suggest that UBE3A limits the hippocampal acute GR response but not in the context of prolonged glucocorticoid exposure. However, continuous CORT exposure did lead to a statistical interaction between AS and CORT treatment in the liver transcriptome, which suggests that UBE3A may regulate GR activity in a tissue-specific manner. The observed changes in the liver transcriptomic response to continuous CORT treatment are in line with AS mice susceptibility to hepatic dysfunction [34].

In conclusion, our current study provides novel insight into the molecular interaction between the UBE3A and GR. We demonstrate that AS mice display reduced circulating corticosterone levels during the circadian trough, and a vastly different transcriptomic response in the hippocampus following acute corticosterone treatment. We believe that our findings contribute to a better understanding of the molecular mechanisms of altered glucocorticoid responsiveness in AS. For future research, the corticosterone treatment in AS mice provides a suitable model to investigate the effectiveness of GR antagonists that could compensate for the altered glucocorticoid sensitivity in AS mice, and such treatment strategies could potentially benefit AS patients that suffer from stress and anxiety symptoms [61,62,63,64,65].

## 4. Materials and Methods

### 4.1. Microarray Assay for Realtime Coregulator-Nuclear Receptor Interaction (MARCoNI)

The GR is a nuclear receptor with two activation function domains (AF1 and AF2), a DNA-binding domain (DBD) and a ligand binding domain (LBD). GR interactions with coregulator motifs via its LBD were assessed in the presence of 1 µM cortisol compared to vehicle (dimethyl-sulfoxide, DMSO), as previously described [66]. A total of 154 short leucine rich (LxxLL) binding motifs representing 64 known nuclear receptor coregulators were attached to the solid phase of the PamChip^®^ array. The GR LBD tagged with Glutathione S-transferase (GST) was overexpressed in HEK 293T cells. The coregulator-derived motifs were incubated with the HEK 293T cell lysates containing GR LBD-GST, the ligand (DMSO or 1 µM cortisol), and a GST-specific antibody coupled to a fluorophore. The interaction between GR LBD and the coregulator motifs was assessed by detection of the fluorophore immunofluorescent signal. For all coregulator motifs, each treatment condition had four replicates from which the mean and the standard error of the immunofluorescent signal were calculated. The modulation index (MI) of each coregulator motif was calculated as the log10-transformed ratio of the mean signal in the presence of cortisol to the mean signal in the presence of DMSO. The standardized MARCoNI assay numbered the LxxLL domain amino acids based on the sequence of the human isoform 2 of UBE3A, however this isoform is lowly expressed [39]. Therefore, the sequence of isoform 3 of UBE3A, in concordance with the MANE annotation [67], was further used for amino acid numbering in the current study.

### 4.2. Cell Culture

#### 4.2.1. GR Activity 

Wildtype (WT) and UBE3A knock-out (UBE3A^KO^) HEK 293T cells [17] were seeded 80,000 cells/well in 24-well plates in culture medium DMEM (1X) + GlutaMAX™ supplemented with 10% Charcoal-stripped fetal bovine serum (FBS), penicillin and streptomycin. The cells were transfected after 24 h using the FuGENE^®^HD Transfection Reagent (E2311, Promega Corporation, Madison, WI, USA) according to the manufacturer’s instructions. Per well, 10 ng human GR plasmid, 25 ng tyrosine aminotransferase (*Tat*) GR response element coupled to the firefly luciferase (TAT1—GR-dependent reporter gene) [68], 1 ng CMV-renilla (control reporter gene) and 264 ng pcDNA were transfected. One day later, the cells were treated with 0.1% DMSO as vehicle or 0.1–1000 nM cortisol. After 24 h, the medium was removed, the cells were washed once with phosphate-buffered saline (PBS) and lysed to measure GR-dependent luciferase activity using the Dual-Luciferase^®^ Reporter Assay (E1910, Promega Corporation, Madison, WI, USA). The luminescent signal was measured with SpectraMax^®^iD3 (Molecular Devices, San Jose, CA, USA), using SoftMax Pro 7.0.2. Each treatment condition had four replicates and the experiment was performed twice. All data are expressed as the ratio between the reporter gene and the control reporter gene luminescent signal.

#### 4.2.2. GR Stability

UBE3A^KO^ HEK 293T cells were seeded at a density of 150,000 cells/well in 12-well plates and cultured in DMEM (1X) + GlutaMAX^TM^ (Gibco^TM^, 10566016, ThermoFisher Scientific) supplemented with 10% fetal calf serum (FCS), penicillin, and streptomycin. After 24 h, the cells were transfected with plasmids containing the human GR (750 ng) in combination with different human UBE3A isoform–1 variants (UBE3A^WT^, UBE3A^LD^, UBE3A^dAZUL^) (750 ng) using polyethylenimine (PEI) in a ratio DNA:PEI of 1:3. After 48 h, the cells were treated for eight hours with either vehicle (DMSO) or 1 µM cortisol, alone or in combination with 10 µM MG132 (proteasome inhibitor) or 70 µg/mL cycloheximide (CHX—de novo protein synthesis inhibitor). Hereafter the cells were harvested and snap-frozen in liquid nitrogen until further analysis. Cell pellets were lysed in lysis buffer (250 mM sucrose, 20 mM HEPES [pH 7.2], 1 mM MgCl2, 10 U/mL Benzonase), supplemented with the c0mplete Protease Inhibitor Cocktail (11836145001, Roche, Basel, Switzerland). Protein concentrations were determined using BCA Protein Assay Kit (23225, Pierce™ Thermo Fisher Scientific Inc., Waltham, MA, USA) and a total of 10 µg protein per sample was loaded and run on a 26-well 4–15% Tris-Glycine gel (5671085, Bio-Rad Laboratories, Hercules, CA, USA). The separated proteins were transferred to a 0.2 mm nitrocellulose membrane (170-4159, Bio-Rad Laboratories, Hercules, U.S.A.) and blocked in TBST (10 mM Tris-HCl [pH 8.0], 150 mM NaCl, 0.1% Tween-20 (P1379, Sigma-Aldrich, St. Louis, MO, USA), supplemented with 5% (w/v) skim-milk powder (70166-500G, Sigma-Aldrich, St. Louis, MO, USA) for 30 min at room temperature. Membranes were briefly rinsed with TBST, and incubated with primary antibodies overnight at 4 °C. The day after, the membranes were briefly washed in TBST, before incubation with secondary antibodies for an hour at room temperature. Finally, the membranes were washed three times with TBST and three times with TBS and analyzed by measuring enhanced chemiluminescence or infrared fluorescence (Li-Cor Biosciences, Lincoln, NE, USA). The primary and secondary antibodies listed in Table 1 were used:

Protein band intensities were measured using ImageStudioLite (Version. 5.x 4.0 3.1.) and normalized against β-Actin or GAPDH protein expression. The experiment was performed four times with three to four replicates per condition. The average of all replicates was calculated in each assessment giving a total N = 4 for the entire experiment. GR protein levels were expressed as a normalized percentage from the vehicle-treated control group.

### 4.3. Animals

WT C57BL/6J male and female 129sv *Ube3a*^m+/*p*−^ mice (*Ube3a*^tm1Alb^; MGI 2181811) [50], were crossed to generate F1 hybrid WT and *Ube3a*^m−/p+^ (AS) offspring. The *Ube3a*^tm1Alb^ strain resulted from the deletion of the exon 5 of *Ube3a*, leading to an out-of-frame mutation and loss of UBE3A protein expression in neurons. Adult 10–18-week-old male mice were used in the experiments. Mice were housed in conventional cages with a 12:12 h light-dark cycle (7:00 AM lights on and 7:00 PM lights off) and ad libitum access to food and water. All animal experiments were conducted at the Erasmus MC in Rotterdam, in accordance with the European Commission Council Directive 2010/63/EU (CCD approval AVD101002016791).

#### 4.3.1. Acute Corticosterone Exposure

On day 0 of the experiment, blood was withdrawn from WT (*n* = 12) and AS (*n* = 12) mice via tail vein incision, at 8:00 (AM) and 18:00 (PM), to measure baseline circulating corticosterone. Blood samples were kept on ice and centrifuged at 12,000 rpm for five minutes. The plasma was then collected and stored at −20 °C until further processing. On day 5, animals were injected subcutaneously with 3 mg/kg corticosterone (27840, Sigma-Aldrich, St. Louis, MO, USA) or vehicle (5% ethanol in PBS) between 9:00 and 11:00 to mimic a transient increase in corticosterone, as previously described [69,70,71]. Three hours later, animals were sacrificed by cervical dislocation, trunk blood was collected in an EDTA-coated collection tube and kept on ice and centrifuged at 12,000 rpm for five minutes. The plasma was then collected and stored at −20 °C until further processing. The left frontal cortex and the left hippocampus tissues were isolated, snap-frozen in liquid nitrogen and stored later at −80 °C until further processing.

#### 4.3.2. Continuous Corticosterone Exposure

On day 0 of the experiment, WT (*n* = 10) and AS (*n* = 10) mice underwent surgery to subcutaneously implant a corticosterone slow-release pellet (20 mg corticosterone, 80 mg cholesterol) or a vehicle pellet (100 mg cholesterol) to mimic a sub-chronic corticosterone exposure, as previously described [72,73,74]. On day 6, animals were sacrificed by cervical dislocation, and left frontal cortex, left hippocampus and liver tissues were collected, snap-frozen in liquid nitrogen and stored later at −80 °C until further processing.

### 4.4. Corticosterone Biochemical Analysis

To evaluate corticosterone blood levels, plasma samples were analyzed using the Corticosterone HS High Sensitivity EIA kit (AC-15F1, Immunodiagnosticsystems IDS, East Boldon, UK) according to the manufacturer’s instructions. To calculate the delta PM-AM at baseline, for each animal, the AM corticosterone levels (ng/mL) were subtracted from the PM corticosterone levels (ng/mL).

### 4.5. Protein Levels Measurements in Mouse Brain and Liver Tissue

Frozen frontal cortex and liver tissue were lysed with glass beads in RIPA buffer (89900, Pierce™ Thermo Fisher Scientific Inc., Waltham, MA, USA) supplemented with protease and phosphatase inhibitors (A32959, Pierce™ Thermo Fisher Scientific Inc., Waltham, MA, USA) using the homogenizer (Zentrimix 380R, Hettich Benelux B.V, Geldermalsen, The Netherlands) at 4 °C, with 1500 rpm for 40 s. Protein lysates were transferred in a fresh Eppendorf and protein concentrations were determined using a BCA protein assay kit (23225, Pierce™ Thermo Fisher Scientific Inc., Waltham, MA, USA). All protein lysates were diluted in RIPA to the final concentration of 0.8 µg/µL. Protein expression was measured using the Simple Western™ WES system (ProteinSimple Bio-Techne, Minneapolis, MN, USA) with the anti-mouse (DM-002, ProteinSimple Bio-Techne, Minneapolis, MN, USA) or anti-rabbit (DM-001, ProteinSimple Bio-Techne, Minneapolis, MN, USA) detection modules according to the procedure provided by the manufacturer. The primary antibodies listed in Table 2 were used: 

### 4.6. RNA Sequencing (RNA-seq) Analysis of Mouse Hippocampus and Liver Tissue

Total RNA was isolated from frozen hippocampus and liver tissue using the NucleoSpin^®^ RNA kit (740955.50, Macherey-Nagel, Düren, Germany) and RNA quality was assessed using the RNA 6000 Nano kit on the Bioanalyzer (Agilent, Santa Clara, CA, USA). All samples had an appropriate RNA Integrity Number (RIN) over 8 with a 28/18s ratio over 1 and were considered suitable for sequencing. Aliquots of total RNA samples were sent for RNA-seq at BGI Genomics (Tai Po, Hong Kong). Stranded mRNA libraries were constructed, and 100-bp paired end sequencing was performed on the DNBseq platform resulting in over 20 million reads per sample. The BioWDL pipeline was used for reads quality control, alignment, and quantification. Quality control was performed using FastQC and MultiQC. Reads were aligned to mm10 using STAR and gene-read quantification was performed using HTSeq-count (BioWDL v5.0.0, https://biowdl.github.io/RNA-seq/v5.0.0/index.html, accessed on 22 December 2022). Output files were merged into a count matrix as input for differential gene expression analysis. DESeq2 (version 1.30.1) was used for normalization of the data (median of ratio’s method) and identification of differentially expressed genes in R v4.0.2. Differential exon usage analysis was performed in the mouse cohort exposed to continuous corticosterone to validate UBE3A deficiency at the mRNA level in the AS model (Appendix A).

For the differential gene expression analysis, we selected all genes which were expressed in a minimum of three replicates with more than 20 normalized counts for at least one of the treatment groups. This resulted in 14,611 differentially expressed genes for the hippocampus after acute corticosterone exposure, 14,949 genes for the hippocampus after continuous corticosterone exposure, and 11,988 genes for the liver after continuous corticosterone exposure. The sources of variation between groups were identified using the principal component analysis (PCA). The contrasts between groups were analyzed for differential expression in a pair-wise comparison.

### 4.7. Pathway Enrichment Analysis 

The subset of genes contributing to the differential response to acute corticosterone in AS mice underwent gene ontology (GO)-term and pathway enrichment analyses using the STRING database v11.5 (https://string-db.org/, accessed on 22 December 2022) [75]. We selected the top 10 GO-terms for molecular function (GO:MF), and the top 10 pathways detected in REACTOME. The extent of the enrichment was measured by the strength which represents the log10-transformed ratio between i) the number of genes in the current study that are annotated with a term/pathway and ii) the number of genes that were expected to be annotated with this term/pathway in a random network of comparable size. The false discovery rate (FDR) describes how statistically significant the enrichment is, the *p*-values are corrected for multiple testing using the Benjamini-Hochberg approach. Enrichment significance is expressed as the —log10(FDR).

### 4.8. Single-Cell RNA Sequencing in the Adult Mouse Hippocampus

The top 20 downregulated and top 20 upregulated genes associated with the differential response to acute corticosterone in AS mice were selected for single-cell expression analysis in the adult mouse hippocampus, as we previously described [32]. Our published dataset allows the exploration of gene expression in 13 different cell types of the adult mouse hippocampus, including glutamatergic neurons, GABAergic neurons, and non-neuronal cells. The current data was expressed using the dotplot function of Seurat v3.1.5 in R v3.6.1 where the Z-Score (scaled and centered normalized expression) represents the average gene expression within one cell type and the size of the dot indicates the percentage of positive cells.

### 4.9. Statistics

In the MARCoNI assay, statistical significance was determined by a t-test in R v.4.0.2. For the assessment of GR activity in WT and UBE3A^KO^ HEK-293T cells, statistical significance was calculated using the two-way analysis of variance (ANOVA) in GraphPad Prism 9. The EC50 values for both cell lines were determined using GraphPad Prism 9 non-linear fitting and expressed as mean and standard deviation (±SD). The results of GR stability in UBE3A^KO^ HEK-293T cells, mouse plasma corticosterone and tissue protein measurements were corrected for outliers using the Grubb’s method, all results were expressed as mean and standard error of the mean (±SEM) and statistical significance was calculated using the two-way ANOVA or the Mann–Whitney test in GraphPad Prism 9. In the RNA-seq analysis, an FDR-adjusted *p*-value of 0.05 was used as a cut-off to determine differentially expressed genes.

## Figures and Tables

**Figure 1 ijms-24-00303-f001:**
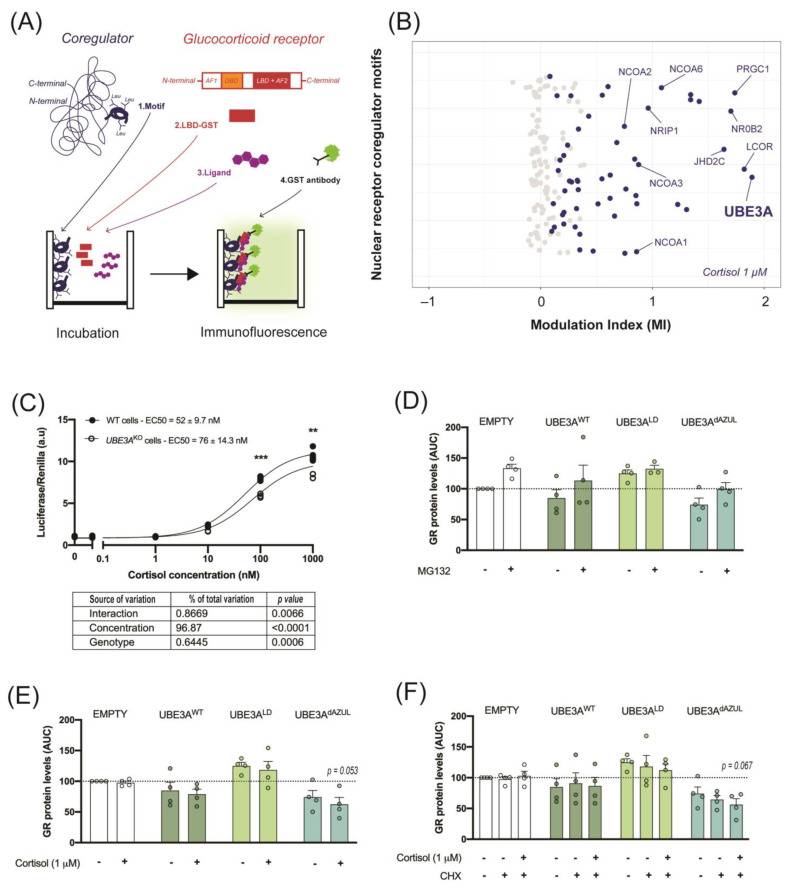
UBE3A interacts with the GR ligand binding domain and slightly alter GR activity and stability. (**A**) Schematic representation of the Microarray Assay for Realtime Coregulator-Nuclear receptor Interaction (MARCoNI) applied to the GR. Short leucine-rich binding motifs corresponding to known nuclear receptor coregulators (1. Motif) were incubated with the GR LBD tagged with Glutathione S-transferase (2. LBD-GST) and a GR ligand (3. Ligand). The interaction between a motif and the GR LBD was detected with a GST-specific antibody coupled to a fluorophore (4. GST antibody). (**B**) Interaction of the GR LBD with 154 short leucine-rich binding motifs of 64 known nuclear receptor coregulators as measured by the modulation index (MI) in the presence of 1 µM cortisol. (**C**) GR activity in response to 0.1 nM to 1µM cortisol in WT or UBE3A knock-out (UBE3A^KO^) HEK 293T cells as measured in a GR-driven luciferase-reporter assay. ** *p* < 0.01; *** *p* < 0.001. (**D**) GR protein levels in UBE3A^KO^ HEK 293T cells after transfection of an empty plasmid (EMPTY), wild-type UBE3A (UBE3A^WT^), ligase-dead UBE3A mutant (UBE3A^LD^), or dAZUL UBE3A mutant (UBE3A^dAZUL^) followed by treatment with vehicle (0.01% DMSO) or proteasome inhibitor MG132; (**E**) vehicle or 1 µM cortisol; (**F**) vehicle, de novo protein synthesis inhibitor cycloheximide (CHX) or CHX and 1 µM cortisol.

**Figure 2 ijms-24-00303-f002:**
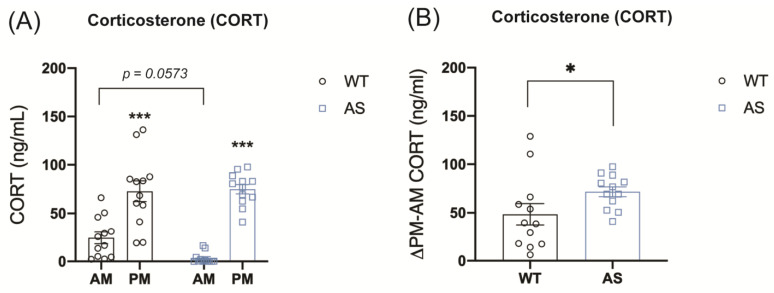
UBE3A deficiency in AS mice alters circulating corticosterone levels. (**A**) Circulating corticosterone levels in WT and AS mice at the beginning of the resting phase (AM) and the active phase (PM) (ng/mL). (**B**) Difference between PM and AM corticosterone plasma levels in WT and AS mice (ng/mL). * *p* < 0.05; *** *p* < 0.001.

**Figure 3 ijms-24-00303-f003:**
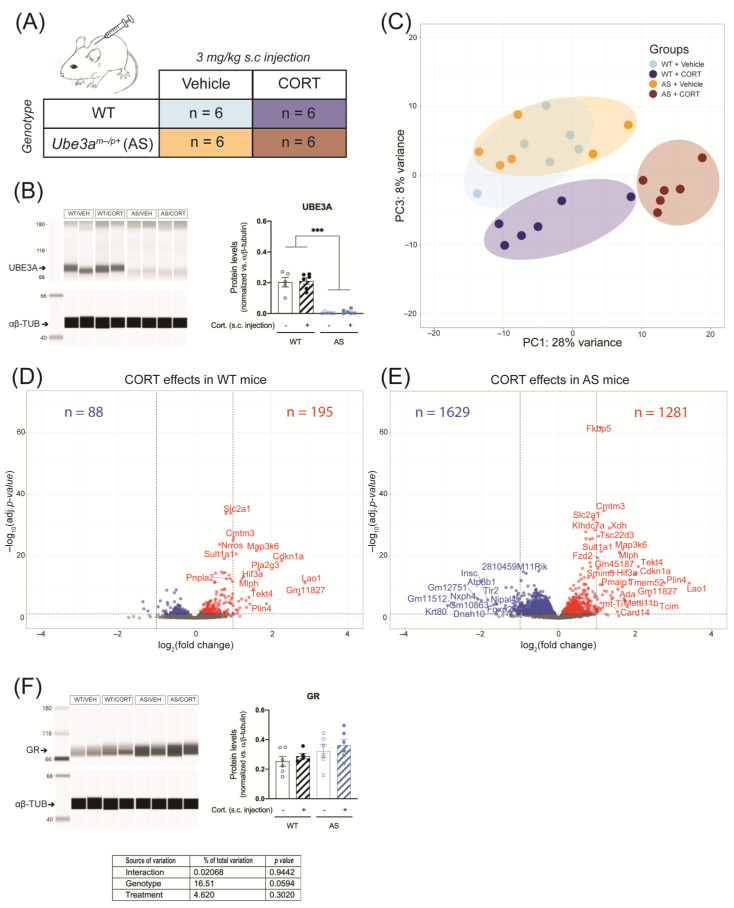
Acute corticosterone exposure strongly alters hippocampal GR signaling in UBE3A-deficient AS mice. (**A**) WT and AS mice were subcutaneously (s.c.) injected with vehicle or 3 mg/kg corticosterone (*n* = 6/group). (**B**) UBE3A protein expression in the frontal cortex of WT and AS mice after acute treatment with vehicle or corticosterone, normalized against αβ-tubulin levels (αβ-TUB). *** *p* < 0.001. (**C**) Analysis of hippocampus RNA-seq data principal sources of variance (PCA). (**D**) Volcano plot representing the transcriptomic effects of acute corticosterone treatment in WT mouse hippocampus. (**E**) Volcano plot representing the transcriptomic effects of acute corticosterone treatment in AS mouse hippocampus. (**F**) GR protein expression in the frontal cortex of WT and AS mice after acute treatment with vehicle or corticosterone, normalized against αβ-tubulin levels (αβ-TUB).

**Figure 4 ijms-24-00303-f004:**
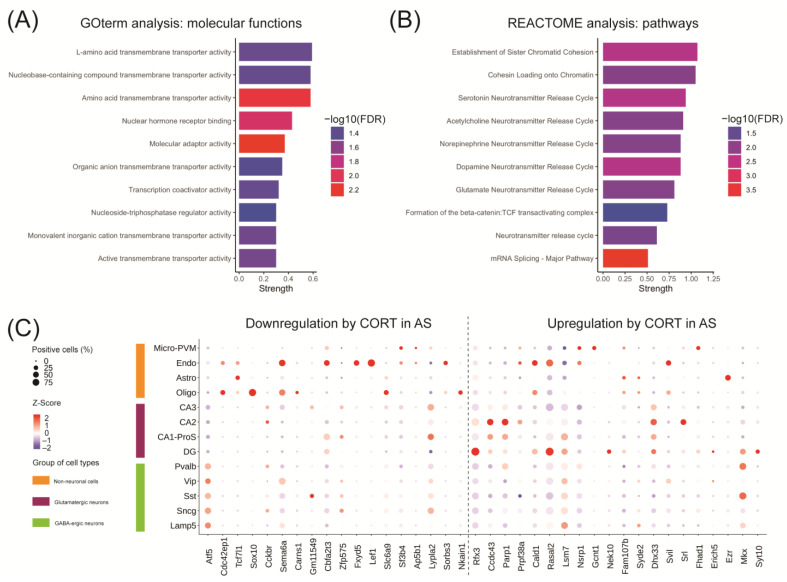
Acute corticosterone exposure in the hippocampus of UBE3A-deficient AS mice influences genes associated with transcription activity and neurotransmitter signaling and are heterogeneously expressed in hippocampal cell types. (**A**,**B**) Top 10 terms identified in the (**A**) gene ontology analysis of molecular functions and (**B**) REACTOME pathway enrichment analysis associated with the 1208 genes differentially altered by acute corticosterone treatment in AS mouse hippocampus. The terms were ranked by strength which represents the log10-transformed ratio between the genes annotated with the term to the number of genes expected to be annotated with this term in a random network of comparable size. Enrichment significance is expressed as the —log10(FDR). (**C**) Top 40 genes differentially altered by acute corticosterone treatment in AS mouse hippocampus expression throughout mouse hippocampal cell types. The size of the dots represents the percentage of cells positive for the gene of interest while the colored Z-Score represents the normalized average expression of the gene of interest within one cell type. Cell types were categorized as: non-neuronal cells, glutamatergic neurons, and GABA-ergic neurons. Glutamatergic neurons were separated based on hippocampal sub-regions: the dentate gyrus (DG) and the cornus ammoni regions (CA1-ProS, CA2 and CA3). Genes were divided according to their direction of regulation after acute corticosterone (CORT) treatment in AS.

**Figure 5 ijms-24-00303-f005:**
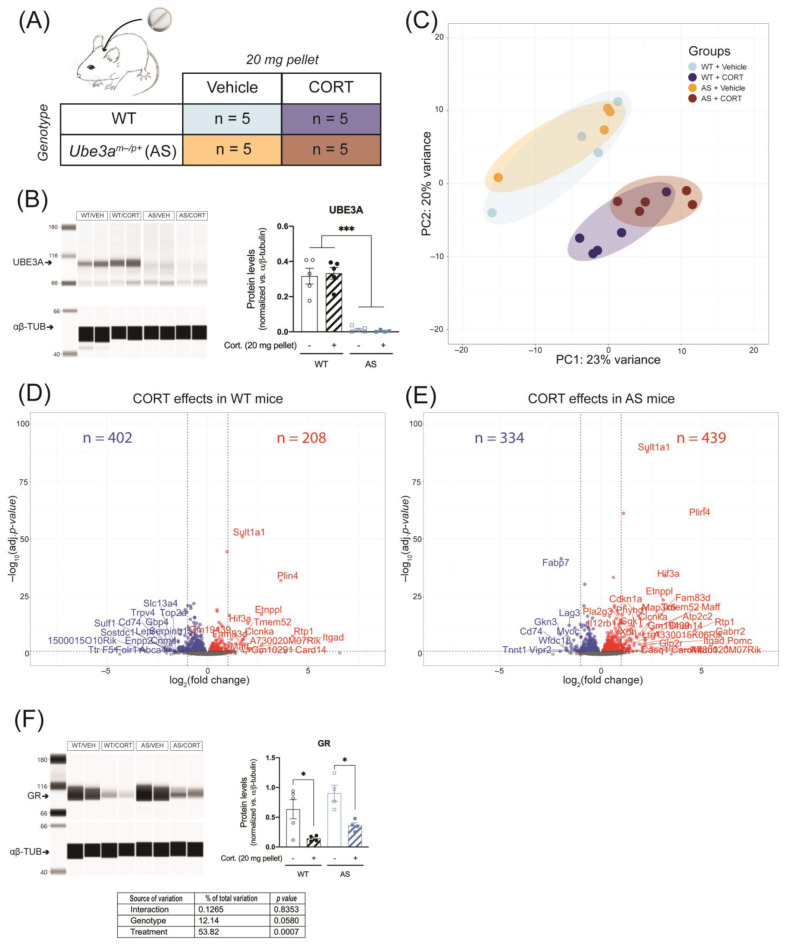
Continuous corticosterone exposure does not differentially alter hippocampal GR signaling in UBE3A-deficient AS mice. (**A**) WT and AS mice were subcutaneously implanted with a vehicle pellet or a slow-release corticosterone (20 mg) pellet for five days (*n* = 5/group). (**B**) UBE3A protein expression in the frontal cortex of WT and AS mice after continuous treatment with vehicle or corticosterone, normalized against αβ-tubulin levels (αβ-TUB). *** *p* < 0.001. (**C**) Analysis of hippocampus RNA-seq data principal sources of variance (PCA). (**D**) Volcano plot representing the transcriptomic effects of continuous corticosterone treatment in WT mouse hippocampus. (**E**) Volcano plot representing the transcriptomic effects of continuous corticosterone treatment in AS mouse hippocampus. (**F**) GR protein expression in the frontal cortex of WT and AS mice after continuous treatment with vehicle or corticosterone, normalized against αβ-tubulin levels (αβ-TUB). * *p* < 0.05.

**Figure 6 ijms-24-00303-f006:**
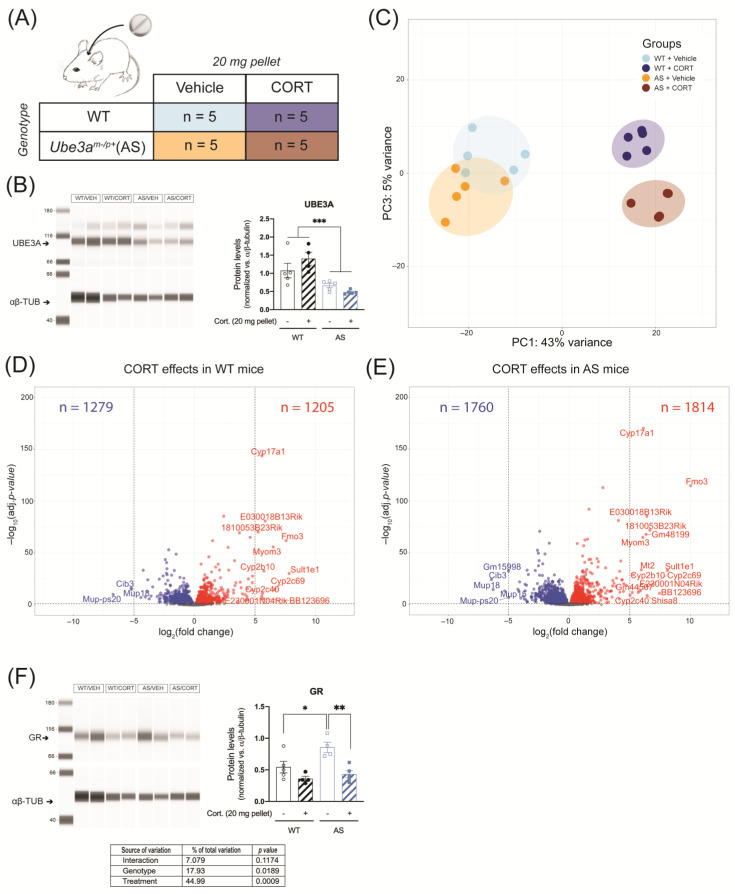
Continuous corticosterone exposure slightly alters liver GR signaling in UBE3A-deficient AS mice. (**A**) WT and AS mice were subcutaneously implanted with a vehicle pellet or a slow-release corticosterone (20 mg) pellet for five days (*n* = 5/group). (**B**) UBE3A protein expression in the liver of WT and AS mice after continuous treatment with vehicle or corticosterone, normalized against αβ-tubulin levels (αβ-TUB). *** *p* < 0.001.(**C**) Analysis of liver RNA-seq data principal sources of variance (PCA). (**D**) Volcano plot representing the transcriptomic effects of continuous corticosterone treatment in WT mouse liver. (**E**) Volcano plot representing the transcriptomic effects of continuous corticosterone treatment in AS mouse liver. (**F**) GR protein expression in the liver of WT and AS mice after continuous treatment with vehicle or corticosterone, normalized against αβ-tubulin levels (αβ-TUB). * *p* < 0.05; ** *p* < 0.01.

**Table 1 ijms-24-00303-t001:** List of antibodies used to assess GR stability in *UBE3A*^KO^ HEK 293T cells.

Antibody Target	Antibody Species	Supplier	Product Number	Dilution
Primary Antibodies				
Anti-GR	Rabbit	Cell Signaling Technology, Danvers, MA, USA	12041S	1:1000
Anti-β-Actin	Mouse	Chemicon Sigma-Aldrich, St. Louis, MO, USA	MAB1501R	1:20,000
Anti-GAPDH	Rabbit	Cell Signaling Technology, Danvers, MA, USA	2118S	1:1,000
Secondary Antibodies				
Anti-mouse	Goat	LI-COR, Lincoln, NE, USA	926-32210	1:15,000
Anti-rabbit	Goat	LI-COR, Lincoln, NE, USA	926-68071	1:15,000

**Table 2 ijms-24-00303-t002:** List of primary antibodies used to assess GR and UBE3A protein levels in animal tissues.

Antibody Target	Antibody Species	Supplier	Product Number	Dilution
Anti-GR	Rabbit	Cell Signaling Technology, Danvers, MA, USA	12041S	1:20
Anti-UBE3A (for brain tissue)	Mouse	Sigma-Aldrich, St. Louis, MO, USA	E8655	1:20
Anti-UBE3A (for liver tissue)	Mouse	Santa Cruz Biotechnology, Dallas, TX, USA	sc-166689	1:20
Anti-αβ-tubulin	Rabbit	Cell Signaling Technology, Danvers, MA, USA	2148S	1:20

## Data Availability

All RNA-seq data have been deposited in NCBI’s Gene Expression Omnibus and are accessible through GEO Series accession number GSE221378.

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
