# Peer review of "The Hippocampal Response to Acute Corticosterone Elevation Is Altered in a Mouse Model for Angelman Syndrome"

_ijms, 2022, doi:10.3390/ijms24010303_

Round 1
Reviewer 1 Report
- Aims of the study should be more clearly emphasized at the end of the introduction with particular emphasis on innovation of the research
- the layout of the article does not correspond to the guidelines of the journal. Materials and methods should be after results and discussion.
- Due to the high number of abbreviation on the text, the list of them at the beginning of the article would increase its readability
- Specifications of reagents should contain the name of company, city, country and catalogue number, not only the name of company such as for example un the line 133,134, 137 and others
- The table with primary and secondary antibodies with their dilutions and specifications would be more readable than listing them in the text (lines 143-148)
- The doses of corticosterone should be justified (line 170 and 179)
- References should not be cited in results chapter (line 264)
- I suggest unifying the article's font (lines 276-279)
- Conclusion is very enigmatic. I suggest to reedit this part of the manuscript. Conclusion should contain the most important observations performed during the study with the emphasis of the novelty of the study and eventually its clinical or diagnostic use (if any).
Author Response
We thank reviewer 1 for his/her valuable feedback. We improved the manuscript with particular intention to the clarity of the introduction and the conclusions. The material and methods section now includes references to previous studies to support our models, and tables that recapitulate the used antibodies with their dilutions.

Reviewer 2 Report
Viho and Punt et al have reported about the changes in hippocampal glucocorticoid receptor signaling after acute corticosterone exposure in mouse model of AS. This paper is well written and the experiments are efficiently performed. However, I have several questions about this manuscript before it can be considered for a publication to this journal.
Below are my comments which need to be answered/explained by the authors,
Major Comments:
1. I am not understanding the justification of putting Figure S2 in the supplementary section. Those results are important for the section 3.1 and can be moved as main figures.
2. Authors have implanted a CORT-releasing pellet subcutaneously and mentioned that as a source of continuous CORT release. Have they measured for any fluctuation in blood in AM and PM? How can they confirm that it was uniform and continuous? Please elaborate on that.
3. Can author discuss about the causal effect of this specific set of gene expressions in disease progression to put more perspective? This can be included in the discussion.
4. What was the rationale behind probing the UBE3A protein content in the frontal cortex tissue, but not in the lysates from hippocampus (Fig 5B and 5F)?
5. There should be a separate section for statistical tests performed throughout this paper, and explicitly summarize what test was applied for each analysis.
Minor Comments:
1. Check the font style in the lines 277-279, 503-511, 590.
2. Regarding HPA axis alteration, what about ACTH changes during AM and PM for AS mice? Any measurement done?
Author Response
We thank reviewer 2 for his/her helpful suggestions. To improve the materials and methods section, we included references to our previous studies using the corticosterone-releasing pellet model in mice. As suggested, we discussed further the relevant gene pathways identified in our model.
